# Effectiveness of Rice Germ Supplementation on Body Composition, Metabolic Parameters, Satiating Capacity, and Amino Acid Profiles in Obese Postmenopausal Women: A Randomized, Controlled Clinical Pilot Trial

**DOI:** 10.3390/nu13020439

**Published:** 2021-01-29

**Authors:** Mariangela Rondanelli, Gabriella Peroni, Attilio Giacosa, Teresa Fazia, Luisa Bernardinelli, Maurizio Naso, Milena Anna Faliva, Alice Tartara, Clara Gasparri, Simone Perna

**Affiliations:** 1IRCCS Mondino Foundation, 27100 Pavia, Italy; mariangela.rondanelli@unipv.it; 2Department of Public Health, Experimental and Forensic Medicine, University of Pavia, 27100 Pavia, Italy; 3Endocrinology and Nutrition Unit, Azienda di Servizi alla Persona ‘‘Istituto Santa Margherita’’, University of Pavia, 27100 Pavia, Italy; mau.na.mn@gmail.com (M.N.); milena.faliva@gmail.com (M.A.F.); alice.tartara01@universitadipavia.it (A.T.); clara.gasparri01@universitadipavia.it (C.G.); 4Department of Gastroenterology and Clinical Nutrition, Policlinico di Monza, via Amati 111, 20900 Monza, Italy; attilio.giacosa@gmail.com; 5Department of Brain and Behavioral Science, University of Pavia, 27100 Pavia, Italy; teresa.fazia01@ateneopv.it (T.F.); luisa.bernardinelli@unipv.it (L.B.); 6Department of Biology, College of Science, University of Bahrain, Sakhir Campus, Sakhir 32038, Bahrain; simoneperna@hotmail.it

**Keywords:** obesity, rice germ, body composition, amino acids

## Abstract

Rice germ (RG) may be a safe and effective dietary supplement for obesity in menopause, considering its high protein content and considerable amounts of essential amino acids, good fatty acids, and fiber. This pilot randomized, blinded, parallel-group, placebo-controlled pilot trial investigated the effectiveness of 4-weeks RG supplementation (25 g twice a day) on body composition, as primary outcome, measured by Dual Energy X-Ray Absorptiometry (DXA), and metabolic parameters, as secondary outcomes, like amino acid profiles and satiating capacity, in obese postmenopausal women following a tailored hypocaloric diet (25–30% less than daily energy requirements). Twenty-seven women were randomly assigned to the supplemented group (14) or placebo group (13). There was a significant interaction between time and group for body mass index (BMI) (*p* < 0.0001), waist (*p* = 0.002) and hip circumferences (*p* = 0.01), total protein (0.008), albumin (0.005), Homeostasis Model Assessment index score (*p* = 0.04), glycine (*p* = 0.002), glutamine (*p* = 0.004), and histidine (*p* = 0.007). Haber’s means over time showed a clearly greater feeling of satiety for the supplemented compared to the placebo group. These findings indicate that RG supplementation in addition to a tailored diet counterbalanced the metabolic changes typical of menopause, with improvements in BMI, body composition, insulin resistance, amino acid profiles, and satiety.

## 1. Introduction

The National Health and Nutrition Examination Survey (NHANES) reported that the age-adjusted prevalence of obesity in 2013–2014 was 40.4% among women in the menopausal age bracket (40 to 59 years) [1]. A Concise Review of the Pathophysiology and Strategies for Management of Weight Gain in Women at Midlife in the Mayo Clinic confirmed that weight gain is common among women in menopause and is accompanied by an increased tendency toward central fat distribution [2]. Visceral fat depots may increase to 15–20% of total body fat, compared with 5–8% in the premenopausal state [3].

This concise review points out that it is imperative that primary care physicians screen midlife women for overweight and obesity and offer appropriate advice and referral in order to counteract this weight gain, particularly visceral fat, which can lead to adverse medical complications such as metabolic health problems, including glycemia disorders or type 2 diabetes mellitus (T2D), dyslipidemia, hypertension [4], and metabolic syndrome [5]; it can also raise the risk of certain cancers, including breast and uterine cancers [6], and of cardiovascular disease (CVD) [7]. This leaves obese postmenopausal women at a higher overall mortality risk, with as much as a four-fold increase in cardiovascular deaths in women with a BMI greater than 29 kg/m^2^ [8]. Hormonal changes in menopause promote higher body fat and abdominal obesity, contributing to the increases in CVD and metabolic risk [9,10], so it is essential to study obesity and establish how best to manage this critical phenomenon in postmenopausal women.

Various studies show that menopause may raise CVD risk by affecting lipid and glucose metabolism, amino acid levels, and body composition. In fact, branched-chain and aromatic amino acids have been linked to insulin resistance [11] and to the risk of future T2D [12], some in an obesity-dependent manner [11].

In a large population of 3204 women aged 40–55 years, postmenopausal women had higher glutamine, glycine, tyrosine, and isoleucine concentrations than premenopausal women, and tyrosine and valine showed suggestive associations, pointing toward a role for menopause in their regulation, although an important limitation of this interesting study is not having considered whether dietary intake or quality differs between pre- and postmenopausal women [13] Therefore, the hypothesis of this study was that menopause may contribute to future metabolic and cardiovascular risk by influencing amino acid concentrations, even if the amino acid changes may possibly be a result of the transitional stages, with no specific role.

It is important to identify new dietary supplements that are safe and effective in these subjects. Rice germ (RG) could be a safe and effective supplement for obesity. Research is seeking uses for rice waste products in the pharmaceutical and nutraceutical fields, considering the potential value of the nutrients they contain [14]. In animal models, rice byproducts could serve as antiobesity supplements, limiting the accumulation of body fat and hepatic lipids and improving the biochemical changes related to diet-induced obesity [15,16].

A recent study reported that RG has a high protein content (18 g per 100 g of edible product) with considerable amounts of essential amino acids (mainly lysine, histidine, and valine), fatty acids (mainly mono- and polyunsaturated), and fiber (7 g per 100 g), as well as high levels of thiamine, pyridoxine, vitamin E, iron, and magnesium [17]. Given its worthwhile nutritional value, therefore, it will be interesting in future studies to explore ways to incorporate RG into supplements to boost nutritional intake for specific populations, such as obese subjects. All of these nutrients (particularly amino acids and fiber) may work synergistically in the management of obesity. A recent randomized clinical trial in athletes found that five weeks of RG supplementation (25 g twice a day) increased mid-arm muscle circumference (MAC) and reduced free fat mass, as measured by dual energy X-ray absorptiometry (DXA) (Lunar Prodigy DXA, GE Healthcare Medical Systems, Madison, WI, USA) [18]. However, despite these characteristics, no study has yet examined the potential beneficial effects of RG supplementation in obese subjects.

Given this background, this study was designed to determine the effectiveness of dietary supplementation with RG on body weight and composition, metabolic parameters, satiating capacity, and amino acid profiles in obese postmenopausal women.

## 2. Materials and Methods

### 2.1. Study Design

This was pilot randomized, double-blinded pilot study. Patients were allocated to either the active treatment or placebo. Women who met the admission criteria and who gave signed informed consent were consecutively assigned a number from a computer-generated randomization list, starting from 1. The number was indicated on a label that identified the treatment and was entered on the case report form (CRF). Subjects were randomized to one of the two arms according to a pattern that ensures balanced treatment assignment (in a 1:1 ratio). Figure 1 illustrates the trial design.

### 2.2. Primary and Secondary Endpoints

As primary endpoints, we examined body composition as measured by DXA (fat mass (FM), free fat mass (FFM), visceral adipose tissue (VAT)). As secondary endpoints, we examined the amino acid profiles (aspartic acid, glutamic acid, alanine, arginine, asparagine, cysteine, citrulline, phenylalanine, glycine, glutamine, isoleucine, histidine, methylhistidine, methionine, ornithine, serine, tyrosine, threonine, tryptophan, valine), biochemical metabolic parameters (total serum cholesterol, triacylglycerol, HDL-cholesterol, LDL-cholesterol, apolipoprotein A, apolipoprotein B, complete blood count, electrolytes, glucose, total proteins, prealbumin, lipase, amylase, iron, glucose, uric acid, creatinine, transaminase alanine aminotransferase, aspartate aminotransferase and gamma glutamyl transferase, vitamin D, vitamin B12, folic acid, homocysteine), inflammation as assessed by C-Reactive Protein (CRP), insulin resistance as assessed by Homeostasis Model Assessment (HOMA), satiating capacity as rated with a visual analog scale (Haber scale), and anthropometric measures (weight, body mass index (BMI), calf circumference, waist circumference, arm circumference, hip circumference).

### 2.3. Participants

We examined women consecutively admitted as outpatients at the metabolic rehabilitation division at Santa Margherita Hospital, Azienda di Servizi alla Persona, Department of Public Health, University of Pavia, Pavia, Italy, with these inclusion criteria: obesity (BMI 30–40 kg/m^2^), age 50–65, postmenopausal (time elapsed after 12 consecutive months without menstruation), sedentary, and non-smoking, who did not drink more than six glasses (one glass: 125 mL) of wine a week, did not drink hard liquor (alcohol content at least 20% alcohol by volume), and who agreed not to take part in any other weight loss program.

Before participation, each woman had a complete medical screening, including vital signs, blood tests, urine tests, and a 12-lead electrocardiogram.

Exclusion criteria were evidence of heart, kidney, or liver disease, or any other condition that might influence the results of the study. Women were also excluded if they met the Diagnostic and Statistical Manual-IV (DSM-V) criteria for a current diagnosis of major depressive disorder, as determined by the Structured Clinical Interview for DSM-V Axis 1 Disorders (SCID-1) [19]. Subjects were also excluded if they were taking any medications for weight loss or control of cholesterol and triglycerides. Other exclusion criteria were type 1 diabetes mellitus, irritable bowel disease, celiac disease, chronic pancreatitis, or antibiotic use in the last three months.

The study was conducted in accordance with the principles of the Declaration of Helsinki, good clinical practice, and Italian national regulatory requirements. All procedures were approved by the University of Pavia Ethics Committee (ethics code number 1402/22052019). Written informed consent was obtained from all subjects before enrolment. Data were gathered from the end of January 2018 to the end of December 2019.

### 2.4. Dietary Supplement

The RG and placebo were supplied in vacuum cans weighing 130 g. Once opened, these were stored in a refrigerator (3–4 °C). Small measuring caps were supplied with the cans to indicate the dose to be taken (25 g, twice a day). The RG or placebo was taken every day (25 g in the morning with breakfast and 25 g in the afternoon as snacks) for four weeks. The RG was supplied by the company “Acquerello” (Tenuta Colombara, Livorno Ferraris, Vercelli, Italy). Table 1 shows its nutritional composition.

The control group was given a placebo consisting of an isocaloric wheatgerm-based supplement with the same flavor and appearance as the intervention product. To optimize compliance, instructions were reconfirmed by phone weekly by the same research dietitian.

### 2.5. Biochemical Analysis

Venous blood samples were drawn from the antecubital vein at 8:00 a.m. in patients fasted for 12 h, to determine plasma amino acid concentrations and metabolic parameters. The concentrations of free amino acids in plasma were measured using the AminoQuant II amino acid analyzer based on the HP 1090 HPLC system with fully automated pre-column derivatization using ortho-phthalaldehyde (OPA) and 9-fluorenylmethyl-chloroformate (FMOC) reaction chemistry. Amino acids were detected by measuring UV absorbance at 338 and 262 nm. The procedure was as follows: 2 mL samples of plasma were de-proteinized by adding 500 µL of 0.5 N HCl and centrifuged at 5000× *g* for 10 min at 5 °C, then the supernatant was concentrated to 200 µL under a nitrogen stream and filtered on a 0.45 µm Millipore filter. Aliquots (1 µL each) were automatically transferred to the reaction coil and derived with the reagents listed above. The remaining de-proteinized serum was stored at −20 °C.

Analyses were done in duplicate and the mean of two independent measurements was reported for each amino acid. The average minimum detectable level of amino acid was 3–5 pmol for each microliter of material injected. Amino acid concentrations were expressed as moles per liter.

Clinical chemistry parameters were measured on a Roche Cobas Integra 400 plus analyzer (Roche Diagnostics, Basel, Switzerland) using specially designed commercial kits supplied by the manufacturer. Total serum cholesterol, triacylglycerol, HDL-cholesterol, LDL-cholesterol, apolipoprotein A, apoliprotein B, complete blood count, electrolytes, glucose, vitamin D, vitamin B12, folic acid, total proteins, prealbumin, lipase, amylase, iron, glucose, uric acid, creatinine, transaminase alanine aminotransferase, aspartate aminotransferase, and gamma glutamyl transferase were measured using enzymatic–colorimetric methods. Serum insulin was measured on a Roche Elecsys 2010 analyzer (Roche Diagnostics, Basel, Switzerland) using dedicated commercial electrochemiluminescent immunoassays. Insulin resistance was defined using the Homeostasis Model Assessment (HOMA) with this equation: fasting plasma glucose (mmol/L) times fasting serum insulin (mU/L) divided by 22.5 [20].

C-reactive protein (CRP) was determined by a nephelometric high-sensitivity CRP assay (Dade Behring, Marburg, Germany). Hemochrome was measured using a Coulter automated cell counter MAX-M (Beckman Coulter Inc., Fullerton, CA, USA). Serum homocysteine was measured with an automated fluorescence polarization immunoassay (FPIA). All parameters were recorded at baseline and after four weeks.

### 2.6. Anthropometric Measurements

All measurements were taken in the morning between 09:00 and 10:00. Body weight was recorded with a standardized technique to the nearest 0.1 kg on a precision scale with the participants wearing light clothing and without shoes. Waist measurements were taken at the midpoint between the lowest rib and the top of the hip bone (iliac crest) using a standardized technique [21].

Anthropometric measurements were taken at baseline and after four weeks in both groups. Body weight and height were measured and the BMI was calculated (kg/m^2^). Sagittal abdominal diameter was recorded at L_4–5_ level in the supine position and waist girth was measured. Anthropometric variables were measured by a single investigator [22]

### 2.7. Metabolic Rate

Metabolic rate (kcal/day) was measured using indirect calorimetry with a validated ventilated hood system (TrueOne 2400 metabolic cart, Parvo Medics, Sandy, UT, USA). The flow meter was calibrated each time before each daily measurement and the metabolic cart was calibrated with reference gas. Once steady state had been reached, expired gases were collected for 10 min and used to calculate metabolic rate. During these assessments, metabolic rate was measured after a baseline night of sleep in the laboratory and an overnight fast. Trained technicians monitored subjects and instructed them to keep their eyes open to make sure they remained awake during the tests [23,24].

### 2.8. Body Composition

Body composition parameters, including free fat mass (FFM), fat mass (FM), android fat, and visceral abdominal tissue (VAT), were obtained using dual energy X-ray absorptiometry (DXA) (Lunar Prodigy DXA, GE Healthcare Medical Systems); the in vivo coefficients of variation (CV) were 0.89% and 0.48% for whole body fat (FM) and FFM, respectively, as calculated with the DXA Prodigy enCORE software (version 17; GE Healthcare). The volume of VAT was calculated using a constant correction factor (0.94 g/cm^3^). The software automatically places a quadrilateral box representing the android region, outlined by the iliac crest and with an upper height equivalent to 20% of the distance from the top of the iliac crest to the base of the skull [25].

### 2.9. Satiating Capacity

Satiating capacity was assessed using Haber’s scale (a visual analog scale from), ranging from −10 (extreme hunger—painfully hungry) to +10 (extreme satiety—full to nausea). The subjects indicated their level of agreement on hunger or satiety by pointing to an appropriate place along the graduated visual scale. All the women did the test every day 30 min before dinner [26].

### 2.10. Tolerance of the Experimental Product

Tolerance of the RG was established on the basis of the absence of side effects, i.e., gastrointestinal symptoms such as nausea and diarrhea. Participants were asked about any side effects of the supplements daily by telephone conversation with a registered dietician.

### 2.11. Weight Loss Program and Food Intake

Body weight loss was induced by a low-energy mixed diet (55% carbohydrates, 30% lipids, and 15% proteins) providing 600 kcal less than individually estimated energy requirements based on the basal metabolic rate. The energy content and macronutrient composition of the diets adhered to the nutritional recommendations of the American Diabetes Association [27,28]. These diets were designed to achieve weight losses of 0.5–1 kg per week; this is considered a low-risk intervention [29]. The research dietitian drew up individual diet plans for each subject. To optimize compliance, dietary instructions were reconfirmed each week by the same dietician. Each consultation included a nutritional assessment and weighing.

A three-day weighed food record was done on two weekdays and one weekend day before the study and during the last week of the intervention. Total calories and macronutrients (proteins, carbohydrates, lipids, and fiber) were calculated using a food nutrient database (Rational Diet, Milan, Italy).

### 2.12. Statistical Analysis

Since this was a pilot study and as there was little prior relevant evidence, the sample size depended on the feasibility of recruitment. The minimal detectable effect size for 80% statistical power with an alpha level of 5% required a total of 27 participants, while the two-sided test was Cohen’s d = 0.25.

An independent *t*-test was used to compare means for the two groups and to test the homogeneity of general characteristics. To detect significant pre- and post-treatment changes (time) within and between the two groups, we fitted a linear mixed model (LMM) [30] for each continuous endpoint, with the time, for the supplemented group, and for the interaction of time*group as fixed effects, specifying a random intercept for each subject in the form of one subject to account for the intra-subject correlation produced by the two different measurements on the same patients (54 observations, but only 27 independents). To assess the treatment effect over time, on the Haber satiety scale we fitted a LMM for longitudinal data [31], with the time, group, and the interaction between time and groups as fixed effects, including a random effect in the form of time*subject and an autocorrelation term (corAR1) to take into account intra-subject correlation and temporal effect for sampling or measurement.

All the models were adjusted for age [32]. As an additional approach, we calculated the area under the curve and the coordinates for the supplemented and control groups using the trapezoid rule [33].

Descriptive statistics are reported as the means ± standard deviation (SD). All analyses were done on R 3.5.1 software using the stats and PK packages [34].

## 3. Results

A total of 27 females with a mean age of 61.89 (SD ± 9.03) years were randomly assigned to the supplemented group (14) or placebo group (13). Table 2 reports their age, body weight, height, BMI and basal metabolic rate values at baseline. There were no significant differences between the two groups.

Table 3 and Table 4 report the within-group pre–post differences for each endpoint.

After multiple test correction, the results indicated significant decreases of BMI (*p* < 0.0001), calf circumference (*p* = 0.04), waist circumference (*p* < 0.0001), arm circumference (*p* = 0.02), hip circumference (*p* = 0.0002), weight (*p* < 0.0001), FM (*p* < 0.0001), percentage FM (*p* = 0.004), VAT (*p* = 0.0001), PLT (*p* = 0.001), percentage of lymphocytes, (*p* = 0.04), WBC (*p* = 0.01), total cholesterol (*p* = 0.004), HDL (*p* = 0.01), LDL (*p* = 0.01), glycemia (*p* = 0.007), insulin (*p* = 0.007), HOMA (*p* = 0.004), CRP (*p* = 0.04), blood urea nitrogen (BUN) (*p* = 0.004), GGT (*p* = 0.01), and triglycerides (*p* = 0.004). There were significant increases in total protein (*p* = 0.004), albumin (*p* = 0.0001), aspartic acid (*p* = 0.004), glutamic acid (*p* = 0.004), alanine (*p* = 0.02), asparagine (*p* = 0.004), glycine (*p* < 0.0001), glutamine (*p* = 0.004), histidine (*p* = 0.004), lysine (*p* = 0.02), tyrosine (*p* = 0.02), and valine (*p* = 0.02) in the supplemented group.

The placebo group presented significant decreases of BMI (*p* < 0.0001), calf circumference (*p* = 0.02), waist circumference (*p* = 0.005), hip circumference (*p* = 0.02), FM (*p* = 0.01), percentage FM (*p* = 0.005), PLT (*p* = 0.001), and insulin (*p* = 0.02); and significant increases in MCV (*p* = 0.02) and creatinine (*p* = 0.05). There were no significant differences in amino acids in the placebo group. Table 5 and Table 6 report the between-group differences.

After multiple test corrections, there was a significant interaction between time and group, meaning that the scores over time changed differently in the two groups for BMI (*p* < 0.0001), waist circumference (*p* = 0.002), hip circumference (*p* = 0.01), weight (DEXA) (*p* < 0.0001), MCV (*p* = 0.04), total protein (*p* = 0.008), albumin (*p* = 0.005), HOMA (*p* = 0.04), glycine (*p* = 0.002), glutamine (*p* = 0.004), and histidine (*p* = 0.007). The Pearson’s pairwise correlation coefficient, 95% confidence interval (CI), and *p*-values were then computed to assess the strength of the associations between histidine and glycine and between HOMA and BMI in the two groups at both time points. There was a significant positive correlation between glycine and HOMA at t_0_ in the supplemented group (r = 0.69, *p* = 0.006) (Table 7).

Figure 2 presents Haber’s means over time separately for the two groups.

In the exploratory scatterplots of the data, a Loess smoothing curve and 95% confidence intervals (grey shading) were added for the relationship between Haber’s mean and time in the two groups. The figure shows an improvement in the feeling of satiety in subjects who had received the supplement, moving from negative to positive values, compared to those assigned placebo. LMM indicated a significant time*group interaction (β = 0.05 (95%CI 0.08; 0.02), *p* = 0.0005). In the supplemented group, the feeling of satiety rose significantly with time (β = 0.04 (95%CI 0.02; 0.06), *p* = 0.0002), while the placebo group showed no difference.

To facilitate the identification of changes in the feeling of satiety rated using Haber’s scale, the full observation period of 60 days was divided into six 10-day intervals. Within- and between-group differences in satiation (Haber’s score) between baseline and these six times are reported in Table 8.

Figure 3 shows Haber’s means over time in the two groups for the six intervals only, with a clear improvement of the feeling of satiety for the supplemented group compared to placebo.

We also calculated the area bounded by the curve of the function, the *x*-axis, and the two lines x = 0 and x = 60. For the supplemented group, the area under the curve (AUC) based on the integral between the points using the trapezoidal method was above the *x*-axis and equal to 51.18, indicating a treatment effect; for the placebo group, it was −154.35, i.e., under the *x*-axis. The difference between the two areas was 205.53, with a 95% CI [190.33–220.72], which did not contain zero, meaning there was a significant difference between the two areas; this result is consistent with that given by regression analysis.

Finally, Table 9 shows the macro- and micronutrient intakes at baseline and at the end of the study in the two groups. None of the supplemented group refused the supplement and no side effects were reported during the study.

## 4. Discussion

Obesity has been associated with a higher risk of cardiovascular diseases, cancer, and osteoporosis [35,36,37]. However, although this is well known, the topic has received little attention in postmenopausal women [38], who constitute a high-risk population for these diseases [36,39]. Therefore, there is currently much emphasis on how menopausal women can lose weight [40].

Dietary supplements for weight management have become very popular and a wide variety are available over the counter [41], so it is important to identify supplements that are safe and effective in obese subjects. RG might be a reliable dietary supplement for obesity management. Researchers are seeking ways to use rice waste in the pharmaceutical and nutraceutical fields, considering the potential value of the nutrients it contains [14]. In animal models, rice byproducts can serve as antiobesity supplements, limiting the accumulation of body fat and hepatic lipids and improving the biochemical changes related to diet-induced obesity [15,16].

Rice byproducts appear to be a promising protein source with good biological values and digestibility [42]. Amino acid scores showed that the soluble protein from broken rice was 26.07% higher than casein and 20.43% higher than egg. The scores for soluble protein from defatted bran were 27.03% higher than casein and 30.93% higher than egg. The extracted soluble protein from bran had higher scores than casein and egg:- respectively 75.56% and 85.74% [43]. Therefore, the high protein efficiency ratio of rice, defined as the proportion of protein that contributes to body growth, is well known [44].

This is the first pilot study that provides early evidence that in a group of obese postmenopausal women four weeks of RG supplementation may have positive effects on body composition, as assessed by DXA and compared with placebo. In the supplemented group, after multiple test corrections, the within-group pre–post differences showed significant decreases of FM, percentage FM, and VAT, as assessed by DXA; no such results for body composition were seen in the placebo group. The focus here is on body composition, because BMI is not considered a valid measure of obesity in postmenopausal women [45]. Therefore, our finding of weight loss, with decreases in FM and VAT while preserving FFM, as assessed by DXA, is a very important goal in this category of patients, who are also at risk of sarcopenic obesity [46].

Another noteworthy finding is the substantial improvement in metabolic health, due to significant decreases in VAT, HOMA index, total and LDL cholesterol, triglycerides, and PRC in the group supplemented with RG only. It is still not clear exactly how this supplementation affects metabolic pathways, but we can postulate several mechanisms. First of all, there is the unique nutritional composition of RG [17], particularly its high contents of histidine and lysine. There was a significant interaction between time and group for plasma glycine, glutamine, and histidine. The increases of these amino acids in the blood may have shifted their composition in the plasma of the obese postmenopausal women. Previous studies have reported differences in amino acid composition in the plasma of people with obesity compared to lean individuals, however the perturbations of amino acid concentrations in obesity and the dynamics of these changes after weight loss are not fully understood [47,48,49,50,51]. In addition, associations of amino acid concentrations with menopause have been reported in large population samples (3204 women aged 40–55 years) [13].

Histidine, a precursor of neuronal histamine, has recently been hypothesized to suppress food intake; there have been reports that histidine might suppress appetite and influence body weight through its conversion to neuronal histamine in the hypothalamus of females [52,53,54,55,56]. Moreover, regarding the amino acids and considering the supplemented group, after multiple test corrections the within-group pre–post differences showed a significant increase of lysine; this was not seen in the placebo group.

Lysine, the precursor to metabolic glutamate, can mediate food intake by way of a neuronal and glutamate sensing mechanism through the liver [57,58,59]. In our focus on satiety, Haber’s means over time showed a clear improvement in the feeling of satiety for the supplemented women compared to the placebo group. Therefore, the suppression of the feeling of hunger might be due to the high contents of lysine and histidine in RG. RG is also rich in fiber, another important component that can induce satiety [60].

Also interesting is the potential role of the high contents of vitamins (particularly B1 and B6) and minerals (particularly magnesium) in the RG in supplemented postmenopausal women—it has been postulated that vitamin B1 and magnesium deficiency are under-diagnosed in obesity and may be important in the progress of obesity and obesity-related chronic disease [61]. Regarding pyridoxine, vitamin B6 is involved in energy metabolism and there is evidence that it can influence the inflammatory process in obesity. This vitamin also deserves to be studied more and taken into consideration when managing obesity [62].

Finally, it is worth noting that RG is an inexpensive dietary supplement, as rice is the second leading cereal crop and RG is one of the most abundant byproducts in the rice milling industry.

This study has some limitations. The first is the small sample size—given that this was a pilot study and that there is little prior evidence, the sample size was determined by the feasibility of recruitment. The second limitation regards the subjects studied, which included only obese postmenopausal women and may limit generalization to the general population. Therefore all these findings must be interpreted with caution and further studies are needed with bigger samples from the general population. Another limitation is that we did not measure vitamins B1 and B6 or magnesium in the blood, so we can only speculate on their roles.

Finally, another limitation is that given the study design (RCT), it cannot offer any clear insight into the possible mechanisms underlying the findings, which must, therefore, remain purely hypothetical.

In conclusion, the abundance of nutrients in RG makes it a potential functional food for disease prevention. Our results indicate that dietary RG supplementation gave a beneficial effect, improving body weight, body composition, and the metabolic changes related to obesity in obese postmenopausal women.

## Figures and Tables

**Figure 1 nutrients-13-00439-f001:**
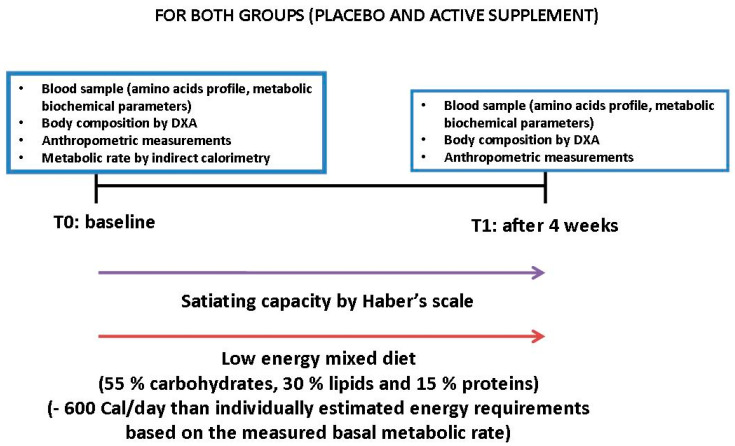
Trial design for the experimental and placebo groups. Abbreviations: Dual energy X-ray absorptiometry: DXA.

**Figure 2 nutrients-13-00439-f002:**
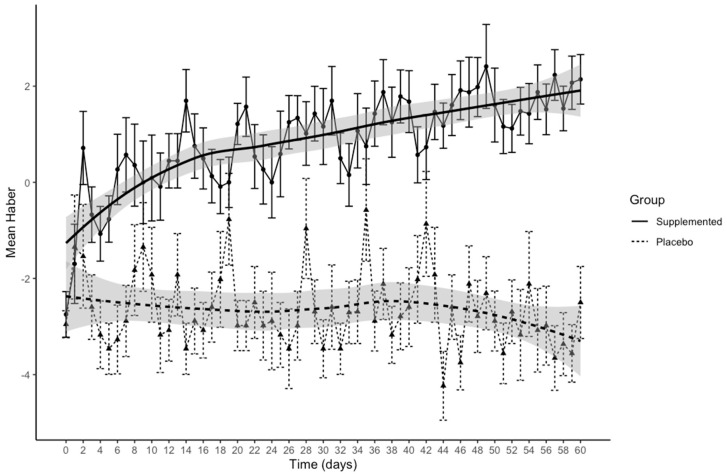
Patterns of changes in Haber’s mean scores versus time for 60 days for satiety in the supplemented and placebo groups, with a Loess smoothing curve and its 95% confidence interval (grey). Error bars indicate the standard error of the mean.

**Figure 3 nutrients-13-00439-f003:**
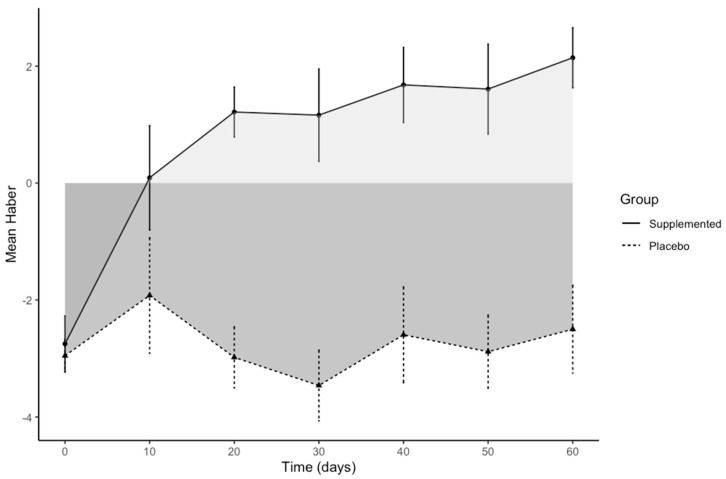
Plot of Haber’s mean scores for satiety against time, considering only the six intervals, shown separately per group. Error bars indicate the standard errors of the mean, while the gray shading indicates the area under the curve for the two groups.

**Table 1 nutrients-13-00439-t001:** Nutritional composition values of rice germ (RG) as the means ± standard deviation.

VARIABLE	Rice Germ
Humidity (g/100 g)	10.53 ± 0.30
Protein (g/100 g)	18.2 ± 1.20
Fats (g/100 g)	17.46 ± 1.91
Dietary fiber (g/100 g)	7 ± 1.50
Ash (g/100 g)	5.65 ± 0.10
Carbohydrates (g/100 g)	41.16 ± 2.70
Energy (kcal/100 g)	409 ± 11
Energy (kJ/100 g)	1711 ± 41
Starch (g/100 g)	25.5 ± 1.22
Vitamin B1 (mg/100 g)	5.8 ± 1.3
Vitamin B6 (mg/100 g)	0.492 ± 0.09
Vitamin E (mg/100 g)	31.9 ± 5.1
Cadmium (mg/kg)	0.0208 ± 0.0025
Iron (mg/100 g)	6.2 ± 1.4
Magnesium (mg/100 g)	347 ± 54.0
Lead (mg/100 g)	n.r.
Sodium (mg/kg)	1.9 ± 0.34
Aspartic acid (mg/100 g)	95.8 ± 17.5
Asparagine (mg/100 g)	74.1 ± 12.3
Glutamic acid (mg/100 g)	130.3 ± 21.5
Alanine (mg/100 g)	41.5 ± 6.6
Arginine (mg/100 g)	115.4 ± 18.6
Cystine (mg/100 g)	<LoQ
Methionine (mg/100 g)	+4.6
Proline (mg/100 g)	21 ± 2.3
Phenylalanine (mg/100 g)	3.3 ± 1.0
Tyrosine (mg/100 g)	+12.2 ± 2.0
Glycine (mg/100 g)	13.9 ± 2.2
Glutamine (mg/100 g)	5.3 ± 1.7
Isoleucine (mg/100 g)	6.4 ± 1.0
Histidine (mg/100 g)	12.7 ± 3.4
Leucine (mg/100 g)	8.5 ± 1.0
Lysine (mg/100 g)	161.2 ± 18.6
Ornithine (mg/100 g)	<LoQ
Serine (mg/100 g)	20.7 ± 0.7
Treonine (mg/100 g)	10.3 ± 3.7
Valine (mg/100 g)	18.3 ± 2.9
Gamma-aminobutyric acid (mg/100 g)	35.4 ± 4.5
IUPAC: 4-aminobutanoic acid	
Alpha-aminobutyric acid (mg/100 g)	<LoQ
IUPAC: 2-aminobutanoic acid-	
Saturated fatty acids (g/100 g)	4.15 ± 0.9
Monounsaturated fatty acids (g/100 g)	5.65 ± 1.1
Polyunsaturated fatty acids (g/100 g)	7.65 ± 42.5

Note: n.r. = not reported; LoQ = limit of quantification.

**Table 2 nutrients-13-00439-t002:** Age, body weight, height, BMI, and basal metabolic rate values at baseline for the supplemented and placebo groups.

	Supplemented Group (14)	Placebo Group (13)	*p*-Value
**Age (y)**			0.91
Mean (±SD)	62.07 (±10.43)	61.69 (±7.66)	
Median (range)	65.5 (44–76)	64 (51–75)	
**Body weight (kg)**			0.32
Mean (±SD)	93.73 (±7.33)	90.08 (±10.69)	
Median (range)	93.6 (84.1–106.6)	90.5 (76.6–119)	
**Height (m)**			0.43
Mean (± SD)	1.61 (±0.06)	1.59 (±0.08)	
Median (range)	1.60 (1.53–1.71)	1.57 (1.48–1.76)	
**BMI (kg/m^2^)**			0.70
Mean (±SD)	36.06 (±1.79)	35.78 (±1.88)	
Median (range)	35.9 (32.5–38.6)	35.4 (33.4–39.9)	
**Basal metabolic rate (kcal/day)**			0.39
Mean (±SD)	1609 (±163.26)	1595.08(±311.69)	
Median (range)	1572 (1439–1956)	1537(1277–2555)	

SD: Standard deviation. BMI: body mass index.

**Table 3 nutrients-13-00439-t003:** Within-group differences for anthropometric, body composition, and biochemical measurements for the supplemented and placebo groups. The estimate of the effect β, 95% confidence interval (CI), and the adjusted *p*-value of the null hypothesis of a null effect are reported.

	Supplemented Group (14)				Placebo Group (13)			
	Mean ± SD (t_0_)	Mean ± SD (t_1_)	β [95%CI]	*p*-Value	Mean ± SD (t_0_)	Mean ± SD (t_1_)	β [95%CI]	*p*-Value
BMI (kg/m^2^)	36.06 ± 1.79	33.29 ± 1.78	−2.77[−3;−2.55]	<0.0001	35.78 ± 1.88	34.51 ± 2.11	−1.28[−1.55;−1]	<0.0001
Calf circumference (cm)	38.89 ± 2.07	38.29 ± 1.85	−0.61[−1.09;−0.12]	0.04	38.88 ± 3.89	38.04 ± 3.58	−0.85[−1.36;−0.33]	0.02
Waist circumference (cm)	115.31 ± 7.75	107.18 ± 6.26	−8.13[−9.83;−6.43]	<0.0001	111.77 ± 9.52	108.63 ± 9.07	−3.14[−4.45;−1.83]	0.005
Arm circumference (cm)	34.57 ± 2.83	33.75 ± 3.09	−0.82[−1.46;−0.18]	0.02	33.41 ± 3.26	32.50 ± 3.04	−0.91[−1.65;−0.18]	0.07
Hip circumference (cm)	118.11 ± 6.14	112.52 ± 6.01	−5.59[−7.36;−3.81]	0.0002	118.54 ± 4.93	116.65 ± 4.13	−1.88[−3.05;−0.72]	0.02
Weight (kg)	93.73 ± 7.33	86.73 ± 7.34	−7[7.83;−6.18]	<0.0001	90.08 ± 10.69	88.05 ± 11.60	−2.03[−2.97;−1.09]	0.008
FFM (g)	50,642.43 ± 6377.05	52,567.29 ± 7288.78	1924.86[509.27;3340.44]	0.02	46,043.62 ± 7691.14	46,218.69 ± 7768.78	175.08[−551.3;901.5]	0.71
FM (g)	40,492.96 ± 4438.19	35,159.21 ± 4841.42	−5333.7[−6417.5;−4249.9]	<0.0001	41,837.77 ± 5970.70	38,650.62 ± 7692.50	−3187.1[−4857.1;−1517.2]	0.01
FM (%)	45.03 ± 5.03	42.46 ± 6.13	−2.57[−3.78;−1.36]	0.004	47.48 ± 4.91	45.88 ± 5.58	−1.61[−2.29;−0.92]	0.005
VAT (g)	2209.71 ± 691.50	1774.36 ± 661.03	−435.36[−559.56;−311.15]	0.0001	1966.61 ± 571.22	1744.15 ± 605.23	−222.46[−452.9;7.98]	0.18
ESR (mm/h)	14.64 ± 8.59	12.43 ± 6.72	−2.21[−6.08;1.65]	0.33	18.54 ± 9.98	19.61 ± 12.60	1.08[−2.58;4.73]	0.65
MCV (fL)	85.91 ± 8.07	85.40 ± 8.24	−0.51[−1.40;0.39]	0.33	88.48 ± 4.65	89.61 ± 4.39	1.14[0.45;1.83]	0.02
PLT (K/uL)	263.5 ± 68.68	227.07 ± 61.40	−36.43[−49.87;−22.98]	0.001	230.54 ± 53.06	207.23 ± 53.24	−23.31[−34.73;−11.88]	0.001
Lymphocytes (K/uL)	2.51 ± 0.76	2.29 ± 0.61	−0.22[−0.44;0.003]	0.09	2.50 ± 0.82	2.14 ± 0.50	−0.35[−0.80;0.10]	0.29
Lymphocytes (%)	35.42 ± 6.49	38.32 ± 6.34	2.90[0.47;5.33]	0.04	35.95 ± 4.69	35.81 ± 8.43	−0.15[−3.89;3.60]	0.96
Prealbumin (mg/dL)	24.43 ± 3.71	26.21 ± 4.54	1.79[−1.03;4.60]	0.28	22.92 ± 4.11	22 ± 2.83	−0.92[−2.90;1.06]	0.49
Total protein (g/dL)	6.75 ± 0.57	7.21 ± 0.80	0.46[0.22;0.70]	0.004	6.59 ± 0.49	6.49 ± 0.41	−0.10[−0.29;0.09]	0.49
Albumin (g/dL)	3.96 ± 0.36	4.33 ± 0.37	0.36[0.26;0.47]	0.0001	3.92 ± 0.24	3.91 ± 0.22	−0.005[−0.16;0.15]	0.96
Hemoglobin (g/dL)	13.48 ± 1.68	13.26 ± 1.51	−0.22[−0.68;0.23]	0.39	13.77 ± 1.06	13.35 ± 0.96	−0.43[−0.75;−0.09]	0.07
Lipase (U/L)	25.36 ± 9.05	25.78 ± 11.37	0.43[−5.66;6.52]	0.90	27.31 ± 26.49	25.08 ± 20.54	−2.23[−6.57;2.11]	0.49
Amylase (U/L)	59.07 ± 20.60	64.14 ± 26.91	5.07[−0.84;10.98]	0.16	52.69 ± 18.15	59.38 ± 19.58	6.69[−1.03;14.42]	0.22
Homocysteine (micromol/L)	17.65 ± 5.26	16.13 ± 4.45	−1.51[−3.09;0.07]	0.11	15.41 ± 3.01	14.73 ± 3.72	−0.69[−2.42;1.05]	0.55
Folate (ng/mL)	12.41 ± 14.09	11.58 ± 9.07	−0.83[−6.55;4.89]	0.79	11.36 ± 8.86	13.82 ± 10.47	2.46[−3.30;8.22]	0.52
Vit B12 (pg/mL)	342.36 ± 88.88	336.71 ± 71.32	−5.64[−37.73;26.45]	0.76	407.92 ± 151.50	394.38 ± 163.00	−13.54[−52.72;25.64]	0.62
Vit D (ng/mL)	25.77 ± 12.55	37.01 ± 12.85	11.24[0.67;21.82]	0.08	24.31 ± 9.91	34.17 ± 14.24	9.85[0.59;19.11]	0.13
Fe (mg/dL)	77.86 ± 25.31	74.79 ± 25.42	−3.07[−12.35;6.20]	0.58	80.54 ± 34.22	82.54 ± 25.00	2[10.68;14.68]	0.81
Na (mmol/L)	139.21 ± 2.52	140.14 ± 2.03	0.93[−0.61;2.47]	0.32	140.92 ± 1.80	140.69 ± 2.39	−0.23[−1.87;1.41]	0.81
K (mmol/L)	4.55 ± 0.41	4.45 ± 0.29	−0.10[−0.32;0.12]	0.42	4.38 ± 0.26	4.41 ± 0.28	0.04[−0.16;0.24]	0.78
Cl (mmol/L)	103.93 ± 3.12	104.86 ± 3.48	0.93 [−0.96;2.81]	0.39	104.31 ± 2.95	104.08 ± 2.96	−0.23[−1.77;1.31]	0.81
Ca (mg/dL)	9.43 ± 0.44	9.47 ± 0.35	0.04[−0.18;0.27]]	0.76	9.46 ± 0.23	9.60 ± 0.41	0.11[−0.13;0.34]	0.49
WBC (K/uL)	7.30 ± 2.43	6.15 ± 2.08	−1.14[−1.86;−0.43]	0.01	6.35 ± 1.40	6.13 ± 1.77	−0.22[−0.97;0.52]	0.65
RBC (M/uL)	4.91 ± 0.78	4.80 ± 0.84	−0.11[−0.28;0.06]	0.28	4.73 ± 0.52	4.58 ± 0.46	−0.15[−0.27;−0.03]	0.07
HCT (%)	41.81 ± 4.81	40.47 ± 4.13	−1.34[−2.68;0.006]	0.09	41.68 ± 3.24	40.91 ± 2.90	−0.77[−1.96;0.42]	0.41
Total cholesterol (mg/dL)	188.57 ± 41.96	159.71 ± 32.50	−28.86[−43.84;−13.87]	0.004	181.69 ± 31.73	167.92 ± 22.47	−13.77[-26.72;−0.82[[]	0.13
HDL-cholesterol (mg/dL)	40.43 ± 6.69	37.07 ± 9.35	−3.36[−5.36;−1.35]	0.01	48.61 ± 11.63	46.15 ± 8.08	−2.46[−7.63;2.70]	0.49
LDL-cholesterol (mg/dL)	123.69 ± 39.22	101.83 ± 27.36	−21.86[−35.64;−8.08]	0.01	108.29 ± 34.68	101.81 ± 27.53	−6.48[−18.27;5.32]	0.49
Glucose in plasma (mg/dL)	95.14 ± 10.72	86.14 ± 12.65	−9[−14.06;−3.93]	0.007	92.38 ± 12.45	95.85 ± 28.70	3.46[−9.98;16.90]	0.70
Insulin (mcU/mL)	15.33 ± 8.39	9.60 ± 4.13	−5.73[−8.92;−2.53]	0.007	15.28 ± 7.65	13.38 ± 6.83	−1.91[−2.98;−0.84]	0.02
HOMA	3.62 ± 2.00	2.10 ± 1.00	−1.52[−2.28;−0.76]	0.004	3.47 ± 1.74	3.28 ± 2.08	−0.18[−0.74;0.37]	0.62
CRP (mg/dL)	0.58 ± 0.44	0.32 ± 0.28	−0.26[−0.47;−0.04]	0.04	0.60 ± 0.47	0.60 ± 0.52	0[−0.37;0.37]	1
Uric acid (mg/dL)	6.48 ± 1.39	6.39 ± 1.38	−0.09[−0.53;0.36]	0.76	6.28 ± 1.10	5.68 ± 1.10	−0.59[−1.43;0.24]	0.36
BUN (mg/dL)	38.00 ± 7.73	35.36 ± 7.15	−2.64[−4.05;−1.24]	0.004	36.77 ± 4.80	35.54 ± 6.38	−1.23[−3.76;1.30]	0.49
Creatinine (mg/dL)	0.84 ± 0.08	0.85 ± 0.08	0.007[−0.03;0.05]	0.76	0.78 ± 0.10	0.81 ± 0.09	0.03[0.008;0.05]	0.05
ApoA (mg/dL)	127.29 ± 19.96	131.07 ± 24.66	3.79[−0.89;8.47]	0.17	125.61 ± 13.31	128.31 ± 13.95	2.69[−2.18;7.56]	0.49
ApoB (mg/dL)	93.79 ± 13.30	87.85 ± 13.91	−5.93[−13.63;1.77]	0.19	92.38 ± 15.36	89.54 ± 14.52	−2.85[−8.26;2.57]	0.49
AST (IU/L)	20.64 ± 6.71	20.79 ± 4.39	0.14[−2.98;3.26]	0.92	21.15 ± 12.66	17.46 ± 3.71	−3.69[−11.31;3.93]	0.49
ALT (IU/L)	26.78 ± 13.02	24.14 ± 6.57	−2.64[−7.82;2.53]	0.38	31.38 ± 32.34	20.77 ± 6.58	−10.61[−30.91;9.68]	0.49
Gamma-GT (U/L)	24.86 ± 9.06	17.64 ± 4.94	−7.21[−11.81;−2.62]	0.01	35.92 ± 39.69	18.23 ± 8.59	−17.69[−40.62;5.23]	0.30
Triglycerides (mg/dL)	122.14 ± 31.32	104.07 ± 23.94	−18.07[−26.77;−9.37]	0.004	118.00 ± 29.68	113.46 ± 27.32	−4.54[−14.46;5.39]	0.49

Data are means ± standard deviation (SD). BMI: body mass index; FFM: free fat mass; FM: fat mass; VAT: visceral adipose tissue; ESR: erythrocyte sedimentation rate; MCV: Mean corpuscular volume; PLT: platelets; WBC: White Blood Cells; RBC: Red Blood Cells; HCT: hematocrit; HDL-cholesterol: High Density Lipoprotein-cholesterol; LDL-cholesterol: Low Density Lipoprotein-cholesterol; HOMA: Homeostasis model assessment; CRP: C- Reactive Protein; BUN: Blood Urea Nitrogen; ApoA: Apolipoprotein A; ApoB: Apolipoprotein B; AST: aspartate transaminase; ALT: alanine transaminase; Gamma-GT: Gamma-Glutamyl transferase.

**Table 4 nutrients-13-00439-t004:** Within-group differences for the 22 plasma amino acids and metabolites in the supplemented and placebo groups. The estimate of the effect β, its 95% confidence interval (CI), and the adjusted *p*-value of the null hypothesis of a null effect are reported.

	Supplemented Group (14)				Placebo Group (13)			
	Mean ± SD (t_0_)	Mean ± SD (t_1_)	β [95%CI]	*p*-Value	Mean ± SD (t_0_)	Mean ± SD (t_1_)	β [95%CI]	*p*-Value
Aspartic acid (μmol/L)	34.93 ± 29.72	45.41 ± 27	10.48[6.04;14.91]	0.004	33.94 ± 20.68	32.50 ± 22.31	1.44[13.30;10.42]	1
Glutamic acid (μmol/L)	56.45 ± 31.72	67.85 ± 33.83	11.40[5.56;17.24]	0.004	54.18 ± 38.50	54.20 ± 20.97	0.01[−8.68;8.72]	1
Alanine (μmol/L)	191.78 ± 73.64	248.12 ± 73.77	56.34[15.33;97.36]	0.02	188.47 ± 56.71	188.09 ± 55.27	−0.38[−33.17;32.41]	1
Arginine (μmol/L)	24.63 ± 14.03	28.46 ± 20.13	3.83[−6.67;14.33]	0.54	27.88 ± 12.31	28.47 ± 17.75	0.59[−3.91;5.08]	1
Asparagine (μmol/L)	83.34 ± 41.20	93.24 ± 40.54	9.90[4.71;15.09]	0.004	82.30 ± 34.47	81.69 ± 35.42	−0.60[−8.52;7.31]	1
Cysteine (μmol/L)	10.39 ± 9.93	11.55 ± 10.49	1.16[−1.76;4.08]	0.54	11.09 ± 3.73	10.48 ± 5.70	−0.60[−2.03;0.82]	0.93
Citrulline (μmol/L)	16.02 ± 10.63	18.56 ± 8.14	2.54[−0.95;6.03]	0.24	14.04 ± 7.45	14.68 ± 6.22	0.63[−0.18;1.45]	0.93
Phenylalanine (μmol/L)	36.76 ± 10.95	39.66 ± 15.21	2.90[−2.29;8.08]	0.35	35.58 ± 4.91	35.31 ± 10.71	−0.27[−4.69;4.16]	1
Glycine (μmol/L)	46.82 ± 16.21	56.11 ± 15.91	9.29[6.44;12.14]	<0.0001	46.03 ± 12.07	47.22 ± 16.77	1.19[−1.33;3.71]	0.93
Glutamine (μmol/L)	13.61 ± 7.76	22.47 ± 9.08	8.86[4.81;12.92]	0.004	13.74 ± 6.45	14.16 ± 6.89	0.42[−1.06;1.90]	0.93
Isoleucine (μmol/L)	34.75 ± 8.39	39.26 ± 12.80	4.51[−2.06;11.08]	0.24	34.19 ± 5.29	33.30 ± 6.66	−0.89[−3.89;2.11]	0.93
Histidine (μmol/L)	73.24 ± 33.71	97.92 ± 29.33	24.69[12.05;37.32]	0.004	72.73 ± 32.21	73.72 ± 26.81	0.99[−3;5]	0.94
Leucine (μmol/L)	59.90 ± 13.38	58.49 ± 18.27	−1.41[−10.47;7.65]	0.78	60.52 ± 9.16	64.52 ± 13.62	3.99[−6.57;14.56]	0.93
Lysine (μmol/L)	53.31 ± 21.34	77.67 ± 27.97	24.37[6.11;42.63]	0.02	51.97 ± 22.96	52.40 ± 21.53	0.42[−7.67;8.52]	1
Methylhistidine (μmol/L)	8.38 ± 9.51	8.14 ± 8.36	−0.24[−7.70;7.21]	0.94	7.44 ± 7.54	7 ± 7.05	−0.45[−1.12;0.22]	0.93
Methionine (μmol/L)	6.80 ± 2.52	9.71 ± 6.74	2.91[−1.10;6.93]	0.24	7.67 ± 2.66	6.87 ± 2.33	−0.80[−3.39;1.78]	0.93
Ornithine (μmol/L)	24.48 ± 11.07	31.98 ± 16.08	7.50[−3.47;18.48]	0.24	29.20 ± 18.56	25.30 ± 19.86	−3.90[−16.98;9.19]	0.93
Serine (μmol/L)	15.47 ± 6.63	21.85 ± 17.27	6.38[−2.17;14.93]	0.24	16.82 ± 7.09	16.34 ± 6.73	−0.48[−3.52;2.55]	1
Tyrosine (μmol/L)	50.4126.10	67.40 ± 23.67	16.98[8.66;25.31]	0.02	48.48 ± 25.92	53.20 ± 27.27	4.72[−3.39;12.84]	0.93
Threonine (μmol/L)	71.81 ± 23.40	67.41 ± 26.69	−4.40[−19.44;10.6]	0.63	70.43 ± 14.83	77.72 ± 20.22	7.29[−11.89;26.46]	0.93
Tryptophan (μmol/L)	25.44 ± 6.35	24.02 ± 10.47	−1.42[−7.06;4.21]	0.65	26.02 ± 6.45	28.06 ± 9.34	2.04[−0.94;5.02]	0.93
Valine (μmol/L)	141.68 ± 41.91	144.42 ± 50.83	18.01[6.56;29.5]]	0.02	140.61 ± 20.38	144.42 ± 37.62	3.81[−7.24;14.87]	0.93

Data are means ± standard deviation (SD).

**Table 5 nutrients-13-00439-t005:** Between-group differences for anthropometric, body composition, and biochemical measurements. Estimates of time*treatment β, 95% confidence intervals (CI), and adjusted *p*-values of the null hypothesis of β = 0 are reported for each endpoint.

Endpoint	Time*Group β [95%CI]	*p*-Value
BMI (kg/m^2^)	1.49[1.16;1.83]	<0.0001
Calf circumference (cm)	−0.24[−0.91;0.43]	0.61
Waist circumference (cm)	4.99[2.93;7.05]	0.002
Arm circumference (cm)	−0.09[−1.01;0.83]	0.84
Hip circumference (cm)	3.70[1.65;5.75]	0.01
Weight (kg)	4.97[3.78;6.16]	<0.0001
FFM (g)	−1749.78[−3300.05;−199.51]	0.14
FM (g)	2146.56[286.58;4006.54]	0.13
FM (%)	0.96[−0.39;2.32]	0.36
VAT (g)	212.90[−30.14;455.93]	0.26
ESR (mm/h)	3.29[−1.78;8.36]	0.43
MCV (fl)	1.64[0.56;2.73]	0.04
PLT (K/uL)	13.12[−3.77;30.01]	0.32
Lymphocytes (K/uL)	−0.13[−0.60;0.33]	0.67
Lymphocytes (%)	−3.05[−7.22;1.13]	0.35
Sarcopenia (kg/m^2^)	0.80[−0.02;1.62]	0.22
Prealbumin (mg)	−2.71[−6.03;0.61]	0.30
Total protein (g/dL)	−0.56[−0.85;−0.27]	0.008
Albumin (g/dL)	−0.37[−0.54;−0.20]	0.005
Hemoglobin (g/dL)	−0.20[−0.74;0.34]	0.61
Lipase (U/L)	−2.66[−9.87;4.55]	0.61
Amylase (U/L)	1.62[−7.52;10.77]	0.77
Homocysteine (micromol/L)	0.83[−1.40;3.05]	0.61
Folate (ng/mL)	3.29[−4.43;11]	0.60
Vit B12 (pg/mL)	−7.90[−55.65;39.86]	0.77
Vit D (ng/mL)	−1.39[−15.54;12.77]	0.84
Fe (mg/dL)	5.07[−9.68;19.82]	0.61
Na (mmol/L)	−1.16[−3.29;0.98]	0.53
K (mmol/L)	0.14[−0.15;0.42]	0.53
Cl (mmol/L)	−1.16[−3.50;1.18]	0.53
Ca (mg/dL)	0.06[−0.24;0.37]	0.77
WBC (K/uL)	0.92[−0.06;1.90]	0.22
RBC (M/uL)	−0.04[−0.24;0.16]	0.77
HCT (%)	0.57[−1.15;2.28]	0.61
Total cholesterol (mg/dL)	15.09[−3.86;34.04]	0.31
HDL (mg/dL)	0.90[−4.21;6]	0.77
LDL (mg/dL)	15.38[−1.98;32.74]	0.26
Glucose in plasma (mg/dL)	12.46[−0.75;25.68]	0.22
Insulin (mcU/mL)	3.82[0.50;7.13]	0.14
HOMA	1.34[0.43;2.24]	0.04
CRP (mg/dL)	0.26[−0.14;0.66]	0.44
Uricemia (mg/dL)	−0.51[−1.38;0.37]	0.50
BUN (mg/dL)	1.41[−1.28;4.10]	0.53
Creatinine (mg/dL)	0.02[−0.02;0.07]	0.53
ApoA (mg/dL)	−1.09[−7.50;5.32]	0.77
ApoB (mg/dL)	3.08[−6;12.16]	0.61
AST (IU/L)	−3.83[−11.43;3.76]	0.53
ALT (IU/L)	−7.97[−27.61;11.67]	0.61
GGT (U/L)	−10.48[−31.82;10.86]	0.53
Triglycerides (mg/dL)	13.53[1.05;26.01]	0.14

Data are means ± standard deviation (SD). BMI: body mass index; FFM: free fat mass; FM: fat mass; VAT: visceral adipose tissue; ESR: erythrocyte sedimentation rate; MCV: Mean corpuscular volume; PLT: platelets; WBC: White Blood Cells; RBC: Red Blood Cells; HCT: hematocrit; HDL-cholesterol: High Density Lipoprotein-cholesterol; LDL-cholesterol: Low Density Lipoprotein-cholesterol; HOMA: Homeostasis model assessment; CRP: C- Reactive Protein; BUN: Blood Urea Nitrogen; ApoA: Apolipoprotein A; ApoB: Apolipoprotein B; AST: aspartate transaminase; ALT: alanine transaminase; Gamma-GT: Gamma-Glutamyl transferase.

**Table 6 nutrients-13-00439-t006:** Between-group differences for plasma amino acids and metabolites—estimates (β) of time*group, 95% confidence intervals (CI), and adjusted *p*-values of the null hypothesis of β = 0 for the 22 amino acids.

Endpoint	Time*Group β [95% CI]	*p*-Value
Aspartic acid (μmol/L)	−11.92[−23.56;−0.27]	0.10
Glutamic acid (μmol/L)	−11.38[−21.19;−1.58]	0.07
Alanine (μmol/L)	−56.72[−107.12;−6.32]	0.08
Arginine (μmol/L)	−3.24[−14.42;7.93]	0.58
Asparagine (μmol/L)	−10.50[−19.35;−1.65]	0.07
Cysteine (μmol/L)	−1.77[−4.93;1.40]	0.36
Citrulline (μmol/L)	−1.90[−5.44;1.63]	0.36
Phenylalanine (μmol/L)	−3.16[−9.69;3.36]	0.39
Glycine (μmol/L)	−8.11[−11.74;]	0.002
Glutamine (μmol/L)	−8.45[−12.68;−4.21]	0.004
Isoleucine (μmol/L)	−5.40[−12.46;1.66]	0.22
Histidine (μmol/L)	−23.69[−36.72;−10.66]	0.007
Leucine (μmol/L)	5.40[−7.75;18.55]	0.44
Lysine (μmol/L)	−23.94[−43.47;−4.41]	0.07
Methylhistidine (μmol/L)	−0.20[−8.24;7.83]	0.96
Methionine (μmol/L)	−3.72[−8.34;0.91]	0.22
Ornithine (μmol/L)	−11.40[−27.52;4.72]	0.25
Serine (μmol/L)	−6.86[−15.77;2.05]	0.22
Tyrosine (μmol/L)	−12.26[−23.33;−1.19]	0.08
Threonine (μmol/L)	11.69[−13.20;36.58]	0.39
Tryptophan (μmol/L)	3.46[−2.74;9.67]	0.36
Valine (μmol/L)	−14.20[−29.37;0.96]	0.13

**Table 7 nutrients-13-00439-t007:** Correlations between the two amino acids, histidine and glycine, and between HOMA and BMI in the two groups at both time points.

	Supplemented Group			Placebo Group		
Correlation	r	95%CI	*p*-Value	r	95%CI	*p*-Value
Histidine and BMI (t_0_)	0.09	−0.46;0.59	0.75	0.14	−0.45; 0.64	0.65
Histidine and BMI (t_1_)	0.14	−0.42;0.63	0.62	0.22	−0.37;0.69	0.47
Histidine and HOMA (t_0_)	−0.17	−0.64;0.40	0.56	0.07	−0.50;0.60	0.81
Histidine and HOMA (t_1_)	−0.30	−0.70;0.30	0.35	0.13	−0.45;0.63	0.67
Glycine and BMI (t_0_)	0.08	−0.47;0.59	0.78	0.10	−0.48;0.61	0.75
Glycine and BMI (t_1_)	0.17	−0.39;0.64	0.56	0.26	−0.34;0.71	0.39
Glycine and HOMA (t_0_)	0.69	0.26;0.89	0.006	0.04	−0.52;0.58	0.90
Glycine and HOMA (t_1_)	0.47	−0.07;0.80	0.09	−0.15	−0.65;0.44	0.63

BMI: body mass index; HOMA: Homeostasis model assessment.

**Table 8 nutrients-13-00439-t008:** Within- and between-group differences in satiation (Haber’s score).

	Between-GroupTime*Group			Within-Group					
				Supplemented Group			Placebo Group		
	β	SE	*p*-Value	β	SE	*p*-Value	β	SE	*p*-Value
Time 1 (day 10)	1.81	1.30	0–18	2.84	0.83	0.005	1.03	1.01	0.33
Time 2 (day 20)	3.99	0.87	0.0001	3.96	0.65	<0.0001	−0.03	0.41	0.95
Time 3 (day 30)	4.42	1.16	0.0008	3.91	0.94	0.001	−0.51	0.50	0.32
Time 4 (day 40)	4.07	1.16	0.002	4.43	0.81	0.0001	0.36	0.78	0.66
Time 5 (day 50)	4.29	0.93	0.0001	4.36	0.76	0.0001	0.07	0.51	0.90
Time 6 (day 60)—(end of observation)	4.44	1.07	0.0003	4.89	0.71	<0.0001	0.45	0.77	0.57

**Table 9 nutrients-13-00439-t009:** Macro- and micronutrient intakes.

VARIABLES	Supplemented Group		Placebo Group	
Calories (Kcal)	1573 ± 45.5	1515 ± 42.2	1635 ± 44.3	1594 ± 39.7
Proteins (g)	78.04 ± 0.43	78.2 ± 0.39	78.43 ± 0.45	78.22 ± 0.51
Fats (g)	50.49 ± 0.57	49.65 ± 0.61	50.85 ± 0.55	51.15 ± 0.63
-Saturated fats (g)	11.48 ± 0.54	11.05 ± 0.49	10.89 ± 0.61	12.21 ± 0.49
-Monounsaturated fats (g)	26.67 ± 0.65	25.89 ± 0.63	27.68 ± 0.71	26.44 ± 0.59
-Polyunsaturated fats (g)	6.03 ± 0.36	6.12 ± 0.33	5.73 ± 0.39	5.69 ± 0.42
Carbohydrates (g)	215.18 ± 10.79	201.68 ± 9.89	230.69 ± 11.3	218.69 ± 12.02
Sugars (g)	51.08 ± 4.67	46.5 ± 3.64	48.65 ± 2.91	50.36 ± 5.02
Fiber (g)	30.73 ± 1.08	32.43 ± 1.01	29.42 ± 1.21	30.56 ± 2.01
Cholesterol (mg)	171.4 ± 8.54	158.84 ± 9.01	169.53 ± 7.42	175.01 ± 8.41
Calcium (mg)	771.19 ± 118.75	898.48 ± 114.98	578.46 ± 123.2	782.47 ± 115.4
Sodium (mg)	1561.21 ± 165.02	1698.22 ± 171.32	1395.26 ± 167.29	1752.02 ± 159.42
Iron (mg)	11.72 ± 0.41	11.88± 0.44	11.31 ± 0.39	12.31 ± 0.51
Zinc (mg)	6.09 ± 0.72	5.07 ± 0.89	7.07 ± 0.91	6.26 ± 0.63
Copper (mg)	1.6 ± 0.08	1.69 ± 0.06	1.53 ± 0.10	1.52 ± 0.09
Thiamine (mg)	1.1 ± 0.02	1.13 ± 0.03	1.08 ± 0.1	1.09 ± 0.9
Riboflavin (mg)	1.55 ± 0.16	1.62 ± 0.15	1.28 ± 0.21	1.66 ± 0.19
Vitamin C (mg)	197.75 ± 54	185.39 ± 49	288.74 ± 61	166.75 ± 73
Vitamin A (μg)	525.98 ± 126.81	526.68 ±133.21	321.81 ± 119.43	658.83 ± 131.92

## Data Availability

The data presented in this study are contained within the article.

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
