# Peer review of "Effectiveness of Rice Germ Supplementation on Body Composition, Metabolic Parameters, Satiating Capacity, and Amino Acid Profiles in Obese Postmenopausal Women: A Randomized, Controlled Clinical Pilot Trial"

_nutrients, 2021, doi:10.3390/nu13020439_

Round 1

Reviewer 1 Report

The manuscript by Rondanelli et al. entitled "Effectiveness of rice germ supplementation..."  describes the effects of daily supplementation with nutrient-rich rice germ on numerous anthropometric, biochemical, clinical, and dietary factors in postmenopausal women. There are several items that would benefit from clarification.

Throughout the manuscript, the terms menopausal and post-menopausal are used interchangeably but these terms may have different definitions and consequent hormonal levels may fluctuate accordingly. Could the terms be clarified such that the participants are indeed post-menopausal (elapsed time after 12 consecutive months without menstruation)?  Were any participants peri-menopausal or menopausal? Was the duration post-menopausal (1 year post versus 5 years post) relatively uniform for each participant or would this be a non-issue?

Please include the equation for HOMA (fasting plasma glucose (mmol/l) times fasting serum insulin (mU/l) divided by 22.5).

It is stated (page 2, paragraph 3) that, "In a large population..." postmenopausal women had higher amino acid concentrations than pre-menopausal women pointing toward a role for menopause in their regulation. Was the dietary intake and/or dietary quality different between these two groups, which might explain the results? Is it possible that this observation (amino acid changes) is a result of the transitional stages with no specific role?

In section 2.2, could the Haber score be generally defined at first mention as a "visual analog scale" with further definition retained as stated on page 6 (section 2.9)?

In section 2.3, it is stated that participants aged 50-65 years with BMIs of 30-40 kg/m2 were enrolled. There is a correlation of increasing hyperlipidemia with age and occurring particularly in obese individuals, which may also be associated with inflammation. It is stated that subjects were excluded if taking medications for hyperlipidemia (triglycerides, cholesterol) or for inflammation. Why was this cohort medication-free? Also, in section 2.3. it is stated that participants were menopausal. Should this be post-menopausal as stated in the manuscript title? 

Table 1 displays the nutritional composition of rice germ. Were the analyses conducted in multiplets such that error bars could be included?  

In Table 5, the units for BMI under the "Endpoint" column are g/m2. Is this correct?

Reviewer 2 Report

The present study investigates the effectiveness of rice germ supplmentation on body composition, metabolic parameters and satiating capacity in obese post-menopausal women. In this mono-centric double-blind study 27 menopausal women with inclusion criteria were randomized between control (placebo, #=13) and experimental group (supplemented, #=14). THe experimental group was supplemented for 4 weeks with rice germ, and various metabolic and hematic parameters were assessed at baseline and at the end of the supplemental period. The results reported in the study indicate that the administration of rece germ in addition to a tailored diet counteracted effectively the metbaolic changes typically associated with menopause, improving BMI, body composition, insulin resistance, amino acid profile and satiety. 
